



# Linking Switzerland's PM$_{10}$ and PM$_{2.5}$ oxidative potential (OP) with emission sources

Stuart K. Grange[1,2], Gaëlle Uzu[3], Samuël Weber[3], Jean-Luc Jaffrezo[3], and Christoph Hueglin[1]

[1]Empa, Swiss Federal Laboratories for Materials Science and Technology, Überlandstrasse 129, 8600 Dübendorf, Switzerland
[2]Wolfson Atmospheric Chemistry Laboratories, University of York, York, YO10 5DD, United Kingdom
[3]Université Grenoble Alpes, IRD, CNRS, Grenoble INP, IGE (Institute of Environmental Geosciences), 38000 Grenoble, France

**Correspondence:** Stuart K. Grange (stuart.grange@empa.ch); Christoph Hueglin (christoph.hueglin@empa.ch); Gaëlle Uzu (gaelle.uzu@ird.fr)

**Abstract.**

Particulate matter (PM) is the air pollutant which causes the greatest deleterious heath effects across the world and PM is routinely monitored within air quality networks where PM mass according to its size, and sometimes number are reported. However, such measurements do not provide information on the biological toxicity of PM. Oxidative potential (OP) is a complementary metric which aims to classify PM in respect to its oxidising ability in lungs and is being increasingly reported due to its assumed relevance concerning human health. Between June, 2018 and May, 2019, an intensive filter-based PM sampling campaign was conducted across Switzerland in five locations which involved the quantification of a large number of PM constituents and OP for both PM$_{10}$ and PM$_{2.5}$. OP was quantified by three assays: ascorbic acid (AA), dithiothreitol (DTT), and dichlorofluorescein (DCFH). OP$_{\mathrm{v}}$ (OP by air volume) was found to be variable in time and space with Bern-Bollwerk, an urban-traffic sampling site having the greatest levels of OP$_{\mathrm{v}}$ among the Swiss sites (especially when considering OP$_{\mathrm{v}}^{\mathrm{AA}}$), with more rural locations such as Payerne experiencing lower OP$_{\mathrm{v}}$. However, urban-background and suburban sites did experience significant OP$_{\mathrm{v}}$ enhancement, as did the rural Magadino-Cadenazzo site during wintertime because of high levels of wood smoke. The mean OP ranges for the sampling period were: 0.4–4.1, 0.6–3.0, and 0.3–0.7 $\mathrm{nmol\,min^{-1}\,m^{-3}}$ for the OP$_{\mathrm{v}}^{\mathrm{AA}}$, OP$_{\mathrm{v}}^{\mathrm{DTT}}$, and OP$_{\mathrm{v}}^{\mathrm{DCFH}}$ respectively. A source allocation method using positive matrix factorisation (PMF) models indicated that although all PM$_{10}$ and PM$_{2.5}$ sources which were identified contributed to OP$_{\mathrm{v}}$ on average, the anthropogenic road traffic and wood combustion sources had the greatest OP$_{\mathrm{m}}$ potency (OP per PM mass). A dimensionality reduction procedure coupled to multiple linear regression modelling consistently identified a handful of metals usually associated with non-exhaust emissions, namely: copper, zinc, iron, tin, antimony and somewhat manganese and cadmium as well as three specific wood burning-sourced organic tracers – levoglucosan, mannosan, and galactosan (or their metal substitutes: rubidium and potassium) were the most important PM components to explain and predict OP$_{\mathrm{v}}$. The combination of a metal and a wood burning specific tracer led to the best performing linear models to explain OP$_{\mathrm{v}}$. Interestingly, within the non-exhaust and wood combustion emission groups, the exact choice of component was not critical, the models simply required a variable to be present to represent the emission source or process. The modelling process also showed that OP$_{\mathrm{v}}^{\mathrm{AA}}$ may be a more specific metric for OP than the other assays employed in this work. This analysis strongly suggests that the anthropogenic and locally emitted





road traffic and wood burning sources should be prioritised, targeted, and controlled to gain the most efficacious decrease in OP$_v$, and presumably biological harm reductions in Switzerland.

# 1 Introduction

## 1.1 Background

Particulate matter (PM) is a major atmospheric pollutant which is very diverse in terms of size, composition, solubility, and
surface area. PM has deleterious effects on human health, reduces visibility, can negatively effect vegetation, and has significant climate effects (Harrison, 2020). With the resolution of the United Nations Human Rights Council stating that access to a clean, healthy, and sustainable environment is a human right (United Nations Human Rights Council, 2021), further understanding of PM and its negative health effects are required. These factors make PM a priority pollutant for management and control and thus, is widely monitored worldwide (World Health Organization, 2021). However, widespread routine PM monitoring
is based primarily on mass within certain size fractions (and to a lesser extent, particle number) which contains no intrinsic information on sources or the potential for biological harm. There is evidence that carbonaceous species and transition metals are more toxic to biological systems than say, inorganic ions (Fang et al., 2017; Daellenbach et al., 2020; Leni et al., 2020). This gives rise to the motivation to define PM in terms of its biological reactivity and toxicity (Zhang et al., 2021).

The quantification of oxidative potential (OP) has the objective of being a "health relevant" metric of ambient PM by
conducting biological toxicological characterisation (Saffari et al., 2014; Borlaza et al., 2018; Bates et al., 2019). OP aims to complement PM mass and number monitoring data and is measured by quantifying the capacity of PM to drive oxidative stress after inhalation in target molecules, generally from the production of reactive oxygen species (ROS) (Delfino et al., 2013; Fang et al., 2016; Yadav and Phuleria, 2020). ROS are free radicals which are formed with molecular oxygen and such compounds can elicit inflammation responses and apoptosis (cell death), via complex triggers and cascades after inhalation, and therefore,
presents a mechanism of biological toxicity caused by ambient PM (Bates et al., 2019). PM toxicity may result in inflammation, respiratory and cardiovascular diseases, cancer, and impediments to neural function (Raaschou-Nielsen et al., 2016; Liu et al., 2018).

A number of toxicological assays have emerged which measure and quantify PM's OP, but to date, no standard definition has been decided on by consensus (Weber et al., 2018; Calas et al., 2019; Weber et al., 2021). However, the ascorbic acid (AA) and
dithiothreitol (DTT) assays have emerged as potential standards to evaluate ROS and OP because of their relatively widespread use (Calas et al., 2017; Shirmohammadi et al., 2017; Yadav and Phuleria, 2020). However, due to the lack of standard operating procedures and calibrations, comparisons of OP measurements conducted by different laboratories is not advised, or at least should be done very cautiously (Janssen et al., 2014; Calas et al., 2018; Molina et al., 2020). OP is usually expressed in one of two ways: OP per volume of air (OP$_v$), or OP per PM mass (OP$_m$). OP$_v$ is usually used for exposure studies because it is a
metric which indicates the amount of OP a given population is exposed to. Contrasting this is OP$_m$ which is a measure of PM's potency to cause OP per a given PM mass unit.





Previous research has indicated that the intrinsic $OP_m$ of PM is highly variable and depends heavily on the constituents which make-up the PM (Daellenbach et al., 2020). Such conclusions indicate that PM from some emission or generation sources have a greater capacity to drive $OP_m$ (OP per µg). Transition metals (for example, iron, copper, and zinc) in particular

have been repeatedly identified with correlation analyses as very potent $OP_m$ drivers, which are generally sourced from road traffic, specifically non-exhaust emissions from tyre, brake, and road wear (Fang et al., 2016; Liu et al., 2018; Bates et al., 2019; Taghvaee et al., 2019; Gao et al., 2020). Primary and secondary organic aerosol have also been identified as a potent driver of $OP_m$ by some (Samake et al., 2017; Samaké et al., 2020), but because of the vast range of organics which can exist in the atmosphere, specific compounds have yet to be identified as the primary cause. Conversely, inorganic PM sources such as

nitrate- and sulfate-rich sources as well as mineral dust have generally been found to have low $OP_m$ (Daellenbach et al., 2020; Weber et al., 2021). This gives rise to a situation when investigating PM at a regional scale, the total mass distribution can be rather uniform, but OP is spatially highly heterogeneous. This is because of the large contributions of inorganic compounds to mass, and importance of very potent, but irregularly emitted constituents such as some metals near transport corridors and organics sourced from wood burning activities in specific communities, for example.

## 1.2  PM in Switzerland

Switzerland's ambient $PM_{10}$ and $PM_{2.5}$ concentrations have progressively decreased since the mid-1990s after widespread monitoring began (Barmpadimos et al., 2011; Gianini et al., 2012; Grange et al., 2018; Hüglin and Grange, 2021). There is a strong site type gradient in Switzerland where rural locations are less polluted with PM when compared to roadside locations. However, woodburning remains popular in some locations, especially south of the Alps, and this can significantly elevate

wintertime PM concentrations in these environments (Sandradewi et al., 2008; Grange et al., 2020). Based on recent intensive measurements, non-exhaust emissions from road vehicles is an emerging issue in Switzerland's urban areas. Brake wear, tyre wear, road wear, and resuspension of road dust have been shown to be important components of Switzerland's urban PM load (Hüglin and Grange, 2021; Grange et al., 2021; Rausch et al., 2022). Such emissions are generated by abrasive processes and although there is a tendency of such PM to be in the coarse-mode, these emissions can also significantly enhance fine PM

concentrations. This is a result of the non-exhaust emission pathways generating PM with median diameters of approximately 3 µm and thus, straddle the boundary between course and fine PM (Harrison et al., 2021). Non-exhaust PM is relevant in respect to OP because such particles are usually metal-rich and metals are thought to be very potent constituents for driving OP. Indeed, previous reports of OP in Switzerland demonstrated the importance metals in the PM mix for enhanced OP (Yue et al., 2018; Daellenbach et al., 2020).

## 1.3  Objectives

The primary objective of this study is to describe Switzerland's ambient OP using observations from five sampling sites between 2018 and 2019. Additionally, two sub-objectives are identified: (*i*) to compare Switzerland's $OP_v$ values with other locations where observations which can be robustly compared with are available and, (*ii*) to use dimensionality reduction techniques, explicitly, positive matrix factorisation (PMF) receptor models, random forest, and multiple linear regression models to identify



what PM emission sources and components are most likely responsible for elevated OP ($OP_v$ and $OP_m$). The implications of Switzerland's $OP_v$ patterns and the identification of PM sources and constituents will be discussed in respect to PM and $OP_v$ management.

## 2 Methods

### 2.1 Sampling sites

Daily PM filter samples were taken at five sampling sites across Switzerland (Table 1; Figure 1) between June, 2018 and May, 2019. The five monitoring sites used for the PM sampling are included in Switzerland's national air quality monitoring network; NABEL (Federal Office for the Environment, 2021). These established sites are used for compliance or regulatory monitoring and have long-term time series available for most common pollutants (Bundesamt für Umwelt, 2021). The sampling sites are located in different environments, ranging from rural, to urban-traffic surrounds. One site, Magadino-Cadenazzo, is located south of the Alps while the other four are located on the Swiss Plateau.

**Table 1.** Basic information for the five monitoring sites in Switzerland which were used for oxidative potential PM measurements.

| Site | Site name | Local ID | Canton | Lat. | Long. | Elev. (m) | Site type |
|------|-----------|----------|--------|------|-------|-----------|-----------|
| ch0002r | Payerne | PAY | Vaud | 46.8 | 6.9 | 489 | Rural |
| ch0008a | Basel-Binningen | BAS | Basel-Landschaft | 47.5 | 7.6 | 316 | Suburban |
| ch0010a | Zürich-Kaserne | ZUE | Zürich | 47.4 | 8.5 | 409 | Urban |
| ch0031a | Bern-Bollwerk | BER | Bern/Berne | 47.0 | 7.4 | 536 | Urban-traffic |
| ch0033a | Magadino-Cadenazzo | MAG | Ticino | 46.2 | 8.9 | 203 | Rural |

### 2.2 Data

High-volume $PM_{10}$ and $PM_{2.5}$ quartz filter (Pallflex Tissuquartz 2500QAT-UP) samples were collected using Digitel DA-80H samplers with flow rates of $30\,\mathrm{m^3\,h^{-1}}$. Daily sampling ran continuously from midnight and midnight for a 12-month period between June 1, 2018 and May 31, 2019. However, for the quantification of constituents beyond simple mass, punches from every fourth-days' filters were taken and analysed. Because the sites form part of the NABEL network, routine flow checks and various tests were regularly conducted in accordance to standard operating procedures.

In total, 908 filters were analysed with three OP assays. Eight-hundred and ninety-nine valid samples were reported, the missing samples were due to sampling or laboratory issues. Additional filter punches were used for a collection of other laboratory analyses to quantify other PM constituents such as elemental components (with inductively coupled plasma atomic emission spectrometry (ICP-AES) and inductively coupled plasma mass spectroscopy (ICP-MS)), ions (ion chromatography (IC)), elemental and organic carbon (thermal optical transmission (TOT) EN16909 method using the EUSAAR2 temperature protocol (European Committee for Standardization (CEN), 2017)), and a collection of additional organics (high-performance liquid chromatographic method followed by pulsed amperometric detection (HPLC-PAD)). The details of these additional





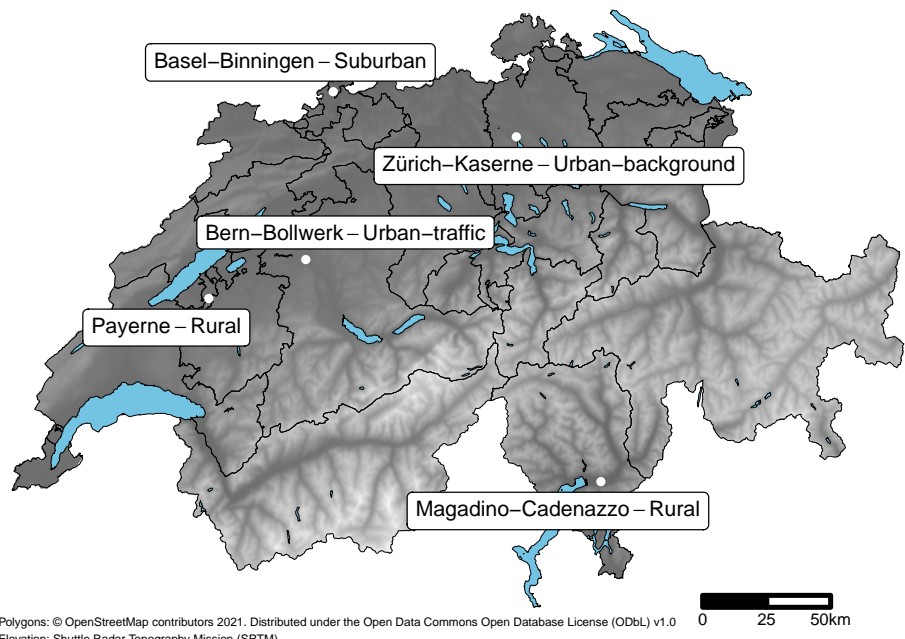

Polygons: © OpenStreetMap contributors 2021. Distributed under the Open Data Commons Open Database License (ODbL) v1.0
Elevation: Shuttle Radar Topography Mission (SRTM)

**Figure 1.** The five sampling sites in Switzerland which were used for oxidative potential PM measurements. The shading indicates the elevation of the terrain and filled blue areas show larger lakes and reservoirs. The cantonal boundaries are displayed as lines.

methods have been reported previously by Hüglin and Grange (2021); Grange et al. (2021) and the latter publication can be
considered companion to this paper.

## 2.3    Oxidative potential assays

OP was analysed with three different assays: ascorbic acid (AA), dithiothreitol (DTT), and dichlorofluorescein (DCFH). These
analyses were conducted at the Institute of Environmental Geosciences, University of Grenoble Alpes, Grenoble, France. The
three different protocols are described in detail in Kelly and Mudway (2003); Cho et al. (2005); Calas et al. (2018, 2019).
All extracts were conducted at iso- and low-mass PM concentration ($25\,\mu\mathrm{g\,L^{-1}}$) to prevent a non-linear measurement effect
(Charrier and Anastasio, 2012; Calas et al., 2018).

The consumption of DTT in the assay was inferred as a measure of the ability of the PM to transfer electrons from DTT to
oxygen, thereby producing reactive oxygen species (ROS). The PM extracts were reacted with DTT, resulting in the consump-
tion of DTT in the solution. The remaining DTT was then titrated with 5,5-dithiobis-(2-nitrobenzoic acid) (DTNB) to produce
a yellow chromophore (5-mercapto-2-nitrobenzoic acid or TNB), which was in direct proportion to the amount of reduced
DTT remaining in solution after the reaction with the PM extract. The consumption of DTT ($\mathrm{nmol\,min^{-1}}$) was determined by
following the TNB absorbance at 412 nm wavelength at 10 min intervals for a total of 30 min of analysis time.





The AA assay relies on one of the main lung antioxidants, ascorbic acid. The consumption of AA $(\mathrm{nmol\,min^{-1}})$ in the assay is inferred as the OP of PM quantified by the transfer of electrons from AA to oxygen. Similar to the DTT assay, the PM
extracts were reacted with AA into a UV-transparent well plate (CELLSTAR, Greiner-Bio). The absorbance was measured at 265 nm using a plate reader (TECAN spectrophotometer Infinite M200 Pro) at 4 min intervals for a total of 30 min of analysis time.

The 2,7-dichlorofluorescin (DCFH) assay is commonly used for detecting intracellular $H_2O_2$ and oxidative stress using a non-fluorescent probe through the formation of a fluorescent product (dichlorofluorescein or DCF) in the presence of ROS
and horseradish peroxidase (HRP). DCF was measured by fluorescence at the excitation and emission wavelengths of 485 and 530 nm, respectively, every 2 min for a total of 30 min of analysis time. The ROS concentration in the sample is calculated in terms of $H_2O_2$ equivalent based on a $H_2O_2$ calibration (100, 200, 300, 400, 500, 1000, and 2000 nmol).

For all assays, the mixtures were injected into a 96-well plates and the absorbance was read from the microplate reader (TECAN spectrophotometer Infinite M200 pro). The well plates were shaken for 3 seconds before each measurement and kept
at $37\,^{\circ}$C. Three laboratory blanks (in Gamble+DPPC) and three positive controls (1,4-napthoquinone at $24.7\,\mu\mathrm{mol\,L^{-1}}$) were included in each plate. The average values of these blanks were then subtracted from the sample measurements of the given plate. Detection limits (DL) were defined as three times the standard deviation of laboratory blank measurements. Uncertainties were estimated thanks to triplicate measurement of the same well.

The three assays have the same objective of determining the amount of oxidative stress an analyte can elicit, but the three
assays have differing sensitivities to various components which form the PM mix and the specific antioxidants within the lung. The three assays have these general characteristics: AA is primarily sensitive to transition metals (Janssen et al., 2014), DTT is the most reported OP assay and is sensitive to organics and to a lesser extent, metals (Janssen et al., 2014; Calas et al., 2019), and DCFH shows a preferential sensitivity to organic compounds. Therefore, the three assays give different perspectives on similar biological processes.

**2.3.1   Units used**

OP can be represented in two forms: OP per PM mass ($OP_m$), or OP per volume of air ($OP_v$). OP per volume of air is a superior unit when representing population exposure and therefore, this unit is mostly used in this analysis. There are three OP assays reported and to differentiate these assays, a superscript notation is used, *i.e.*, $OP_v^{AA}$, $OP_v^{DTT}$, and $OP_v^{DCFH}$ using the $\mathrm{nmol\,min^{-1}\,m^{-3}}$ units.

**2.4   Source apportionment**

Source apportionment for $PM_{10}$ and $PM_{2.5}$ for the five monitoring sites was conducted with the positive matrix factorisation (PMF) receptor model and the multilinear engine (ME-2) algorithm (Paatero and Tapper, 1994; Paatero, 1999). The PMF approach employed was consistent with the SOURCES programme and is informally known as "extended PMF" (Favez et al., 2017; Weber et al., 2019). The EPA PMF 5.0 software tool was used to apply PMF. The specific settings, constraints, and





process of the extended PMF modelling can be found in the accompanying Grange et al. (2021) publication. The PMF input
and output data are also available in a persistent data repository for others' convenience (Grange, 2021a).

The PMF analysis for the particular dataset was challenging because of the existence of fewer than the recommended samples
available (91 compared to the recommended at least 100 (Norris et al., 2014)), low signal to noise ratios for many variables
because of low ambient concentrations, and the inclusion of extra organic species into the PMF models. Despite the many
validation steps conducted, the models had a number of limitations which are discussed fully in the companion paper Grange
et al. (2021) and were considered in the current work.

## 2.5 Linking PM sources to OP

The OP measurements were not included in the PMF modelling process, but these observations required linking to the PMF-
identified sources. To estimate the source contributions to the three OP assays, weighted robust multiple linear regression
(MLR) with an iterative $M$-estimator was used. Conceptually, the OP observations were explained by the PMF-identified
sources, and because linear regression models return coefficients in the dependent variable's response scale, the estimates
of the PMF-identified sources for OP are readily interpreted by investigating the models' slope coefficients ($\beta$). To allow
for evaluation of the models' coefficients uncertainty, the data were bootstrapped 500 times and modelled. Additionally, the
analytical uncertainly was included in the models as weights. The **MASS** R package was used as the interface to the robust
linear regression function (Venables and Ripley, 2002). An example of how this process was conducted can be found in a
public repository (Grange, 2021c).

## 2.6 OP modelling

The filter-based measurement campaign resulted in a large number of elements, ions, and organics to be quantified which
compose Switzerland's PM. To extract the constituents which were the most important for OP, a multiple step process was
conducted to firstly identify the most important constituents which explain OP and secondly, what combination of these con-
stituents resulted in the best statistical models which explained OP values in Switzerland.

The identification of the most important PM constituents to explain OP was conducted with random forest, an ensemble
decision tree machine learning algorithm (Breiman, 2001; Wright and Ziegler, 2017). The entire set of variables available
were used to model OP. The random forests' importances for the included variables were extracted and analysed to reduce the
feature space (Abdulhammed et al., 2019; Reddy et al., 2020). The variables which were consistently identified as the most
important for explanation of $OP_v^{AA}$ and $OP_v^{DTT}$ by random forest were used in further linear modelling work. Therefore, this
dimensionality reduction pre-processing step allowed the dataset to be reduced from over 50 variables to the most important
$\approx 15$ for two OP assays.

The most important variables identified by random forest were used to model OP with robust multiple linear regression
(Venables and Ripley, 2002). Individual models using all combinations of the $\approx 15$ variables with a maximum of five predictors
were created to explain $OP_v$. The intercept term was excluded from the model formulation and over $100\,000$ models were
calculated. An example of how this was achieved is accessible via a public repository (Grange, 2021b). To identify models





which were suitable for further use, three filters were applied to the models. Models with a maximum pairwise variance inflation factor (VIF) for independent variables greater than 2.5 were removed because this suggests multicollinearity among

the independent variables (Jackson et al., 2009). Models which contained negative term estimates were also dropped, as were models with $R^2$ values less than 75 %. These filters resulted in 371 models to be kept for further analysis and the majority (77 %) of these models had two independent variables.

## 3    Results and discussion

### 3.1    Spatial-temporal variation of OP

OP measurements between June, 2018 and May, 2019 at five sampling locations throughout Switzerland demonstrated that $OP_v$ was variable in both time and space. Mean $OP_v$ almost always increased as the sampling location became increasingly urban and Bern-Bollwerk, an urban-traffic site, had the highest levels of $OP_v$ during the sampling period while Payerne, a rural location had the lowest mean $OP_v$ (Figure 2; Table 2). For $OP_v^{AA}$, the $PM_{10}$ means ranged from 0.7 and 4.1 $\mathrm{nmol\,min^{-1}\,m^{-3}}$ and for $PM_{2.5}$, the corresponding range was 0.4 and 1.6 $\mathrm{nmol\,min^{-1}\,m^{-3}}$. $OP_v^{DTT}$ means ranged from 0.8–3.0 and 0.6–

1.1 $\mathrm{nmol\,min^{-1}\,m^{-3}}$ for $PM_{10}$ and $PM_{2.5}$ respectively. $OP_v^{DCFH}$ did not show the same progressive increase across the rural to urban roadside gradient with another rural site, Magadino-Cadenazzo having the highest means (0.7 $\mathrm{nmol\,min^{-1}\,m^{-3}}$ for both $PM_{10}$ and $PM_{2.5}$) while the other four sites were inconsistently ranked for the different PM size fractions and considering the different types of averages (Table 2). The rural-urban-roadside gradient observed for $OP_v^{AA}$ and $OP_v^{DTT}$ was also demonstrated by PM mass and most other individual constituents (the exception was secondary components such as nitrate, sulfate, and

ammonium) which form the Swiss PM mix, and this has been reported previously in a companion paper (Grange et al., 2021).

**Table 2.** Simple summary statistics for three $OP_v$ assays, two PM size fractions, and five sampling sites in Switzerland between June, 2018 and May, 2019. $\bar{x}$ and $M$ represent the mean and median respectively while lower and upper refer to the 2.5 % and 97.5 % quantiles (which contain 95 % of the observations). The summaries have been rounded to one decimal point.

|  |  | $OP_v^{AA}$ |  |  |  | $OP_v^{DTT}$ |  |  |  | $OP_v^{DCFH}$ |  |  |  |
| --- | --- | --- | --- | --- | --- | --- | --- | --- | --- | --- | --- | --- | --- |
| PM | Site | $\bar{x}$ | $M$ | Lower | Upper | $\bar{x}$ | $M$ | Lower | Upper | $\bar{x}$ | $M$ | Lower | Upper |
| $PM_{10}$ | Bern-Bollwerk | 4.1 | 3.8 | 1.3 | 8.4 | 3.0 | 2.6 | 0.9 | 7.6 | 0.4 | 0.3 | 0.0 | 0.9 |
| $PM_{10}$ | Zürich-Kaserne | 1.7 | 1.4 | 0.4 | 4.5 | 1.3 | 1.1 | 0.3 | 2.9 | 0.4 | 0.3 | 0.0 | 1.1 |
| $PM_{10}$ | Basel-Binningen | 1.2 | 0.9 | 0.2 | 3.3 | 0.8 | 0.7 | 0.1 | 1.9 | 0.4 | 0.3 | 0.0 | 1.3 |
| $PM_{10}$ | Magadino-Cadenazzo | 1.7 | 1.2 | 0.2 | 5.5 | 1.0 | 0.8 | 0.1 | 3.3 | 0.7 | 0.4 | 0.1 | 3.2 |
| $PM_{10}$ | Payerne | 0.7 | 0.6 | 0.1 | 2.2 | 0.8 | 0.7 | 0.1 | 2.4 | 0.4 | 0.3 | 0.1 | 1.1 |
| $PM_{2.5}$ | Bern-Bollwerk | 1.6 | 1.4 | 0.5 | 3.8 | 1.1 | 0.9 | 0.2 | 2.2 | 0.5 | 0.3 | 0.1 | 1.1 |
| $PM_{2.5}$ | Zürich-Kaserne | 0.8 | 0.6 | 0.0 | 2.5 | 0.8 | 0.8 | 0.0 | 2.1 | 0.4 | 0.3 | 0.1 | 0.8 |
| $PM_{2.5}$ | Basel-Binningen | 0.7 | 0.4 | 0.1 | 1.9 | 0.6 | 0.4 | 0.1 | 2.0 | 0.4 | 0.2 | 0.0 | 1.3 |
| $PM_{2.5}$ | Magadino-Cadenazzo | 1.2 | 0.6 | 0.0 | 5.0 | 0.7 | 0.5 | 0.1 | 2.0 | 0.7 | 0.3 | 0.1 | 2.7 |
| $PM_{2.5}$ | Payerne | 0.4 | 0.3 | 0.0 | 1.2 | 0.6 | 0.5 | 0.0 | 1.7 | 0.3 | 0.2 | 0.0 | 0.9 |



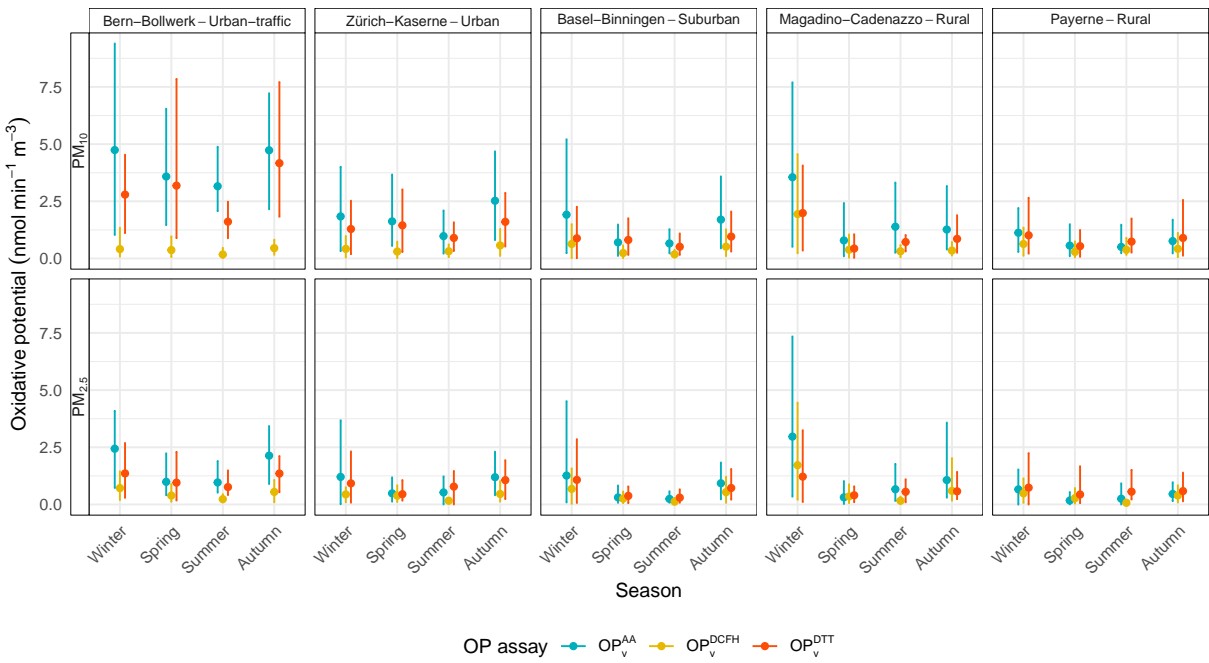

**Figure 2.** Seasonal means of three $OP_v$ assays, two PM size fractions, and five sampling sites in Switzerland between June, 2018 and May, 2019.

Winter and autumn had the highest average OP which is consistent with the common winter situation where primary atmospheric pollutants emissions are higher, and the atmospheric state is less conducive to pollutant transportation and dispersion (Beyrich, 1997; Emeis and Schäfer, 2006). The wintertime OP enhancement was especially clear at Magadino-Cadenazzo, a site known to be heavily burdened by wood smoke during the winter months (Grange et al., 2020). Notably, at Magadino-

Cadenazzo, wintertime $PM_{2.5}$ $OP_v$ was enhanced to nearly the same extent as $PM_{10}$ because of wood burning sourced PM being almost all contained in the fine-mode (Kleeman et al., 1999). Bern-Bollwerk was clearly the most polluted site in respect to $OP_v$ where the two AA and DTT assays remained elevated for all seasons, but mean $OP_v^{DTT}$ was significantly lower during the summer than the other seasons. Another key observation from these aggregations was that $PM_{coarse}$ contained much of the $OP_v$ signal and this is only able to be highlighted because the sampling design included both $PM_{10}$ and $PM_{2.5}$. This point has

two implications, the first is that $PM_{coarse}$ is biologically relevant, and the second is the importance continuing $PM_{10}$ monitoring or sampling because of the aforementioned relevance on human health.

Figure 2 and Table 2 shows large differences among the three assays used to quantify $OP_v$ in this work. The DCFH assay based on a fluorescence method showed much lower levels of structure when compared to the other two, more established AA and DTT assays where the means ranged between 0.3 and 0.7 $nmol\,min^{-1}\,m^{-3}$ (Figure A1). The DCFH assay has a

lower level of sensitivity when compared to the AA and DTT assays, nevertheless the sensitivity of DCFH to organic-rich PM was observed at the southern Magadino-Cadenazzo sampling location where $OP_v$ enhancement was clear during the winter




because of high concentrations of wood burning emissions. The AA assay is primarily sensitive to metals (Janssen et al., 2014), and the Bern-Bollwerk site which is known to experience significant non-exhaust emissions from road traffic, observed high levels of $OP_v^{AA}$ during the sampling period and the mean was $4.1\,\mathrm{nmol\,min^{-1}\,m^{-3}}$. Less severe enhancements were
also observed for the urban and suburban Zürich-Kaserne and Basel-Binningen sites with the AA assay suggesting some metal contamination of these atmospheres too. These observations are consistent with work exploring the urban and roadside increments in Switzerland, and the importance of non-exhaust emissions to these increments (Grange et al., 2021).

### 3.1.1 OP comparison with other locations

The comparison of OP metrics among different locations and sampling durations is problematic due to the lack of standardised
OP laboratory procedures (Calas et al., 2019). Here however, comparisons can be made with many French sites where $OP_v$ has been quantified by the same laboratory with identical analytical approaches. The $OP_v$ of $PM_{2.5}$ has been rarely reported in Europe, and therefore, only $PM_{10}$'s $OP_v$ will be discussed here.

    Based on Weber et al. (2021) which consolidated annual $OP_v$ data for 14 sampling sites across France between 2013 and 2018, Bern-Bollwerk's atmosphere had high levels of $OP_v$ – especially when considering $OP_v^{AA}$ (Table A1). Bern-Bollwerk's
$PM_{10}$ $OP_v^{AA}$ mean of $4.1\,\mathrm{nmol\,min^{-1}\,m^{-3}}$ was substantially higher than all other French sampling locations with the second most polluted location being in Chamonix (site code CHAM), a town in an alpine valley which is topographically confined and where the annual mean $OP_v^{AA}$ was reported as $2.6\,\mathrm{nmol\,min^{-1}\,m^{-3}}$ (between November, 2013 and October, 2014). The four other Swiss sites were within the same range of the reported values for the French locations, however both Zürich-Kaserne and Magadino-Cadenazzo were ranked in the upper half of mean $OP_v^{AA}$ when comparing the 19 sites (14 in France and five
in Switzerland). A map of seasonal and annual $OP_v^{AA}$ and $OP^{DTT}$ means for the closest French sites surrounding Switzerland and the five Swiss sites included in this analysis are shown in Figure 3.

    Bern-Bollwerk also demonstrated high levels of $OP_v^{DTT}$ when compared to the other sampling locations, but for this metric, Chamonix was more polluted than Bern-Bollwerk with means of 4.4 and $2.9\,\mathrm{nmol\,min^{-1}\,m^{-3}}$ respectively (Table A1; Figure 3). The Basel-Binningen, Payerne, and Magadino-Cadenazzo Swiss sites had the lowest $OP_v^{DTT}$ means when consid-
ering the 19 sites which suggests that Switzerland has generally lower levels of $OP_v^{DTT}$ than France, which can be contrasted with $OP_v^{AA}$, where concentrations experienced in Switzerland were similar to those reported across France. However, Bern-Bollwerk was the most polluted sampling location reported in this dataset containing samples analysed by the same laboratories in respect to $OP_v^{AA}$.







**Figure 3.** Seasonal $OP_v^{AA}$ (a) and $OP_v^{DTT}$ (b) $PM_{10}$ means for five Swiss sites included in this analysis and the closest six French sites surrounding Switzerland. Data from the French sampling sites are from Weber et al. (2021).





## 3.2 Linking OP to PM sources

The PMF source apportionment analysis identified eight PM sources in Switzerland: sulfate-rich, nitrate-rich, road traffic, wood combustion, primary biogenic, secondary biogenic, mineral dust, and aged sea salt. All the eight sources were detected for $PM_{10}$ while the primary biogenic, mineral dust, and aged sea salt sources were not identified in the $PM_{2.5}$ fraction indicating that these sources were mostly in the coarse-mode. Full discussion of the PMF results, the limitations, and the sources' characteristics can be found in the companion paper, Grange et al. (2021), however, an outline of the PMF results is briefly given below.

The PMF results indicated that about 50 % of the $PM_{10}$ and $PM_{2.5}$ load in Switzerland was from the three secondary nitrate-rich, sulfate-rich, and aged sea salt sources. Based on the models' factor/source profiles, the former two sources contained a significant amount of organic mass. Generally, the primary and secondary biogenic sources were rather low contributors to average mass concentrations, but they were highly seasonal sources and the secondary biogenic source was more important for $PM_{2.5}$ than $PM_{10}$. The wood burning, mineral dust, and road traffic sources were more enhanced in urban areas, but their

enhancement was highly dependent on the sites' immediate environmental surrounds. Bern-Bollwerk's road traffic source contributed more than a third to both $PM_{10}$ and $PM_{2.5}$, while the wood burning source contributed over 20 % to both PM fractions at Magadino-Cadenazzo, despite also being a source which was inactive for about half of the sampling period.

To investigate the relationship between the activities of the identified main PM sources in Switzerland on its OP, the PMF sources were used in conjunction with the $OP_v$ observations. $OP_v$ was explained using MLR models for each of the five

sites with the identified PMF source contributions as independent variables (in $\mu g\,m^{-3}$). The units of the estimated model coefficients for the PM sources were then in $nmol\,min^{-1}\,\mu g^{-1}$ and interpreted as the intrinsic $OP_m$. This process has been called an 'inversion' by others (Weber et al., 2021; Borlaza et al., 2021) and was conducted 500 times with bootstrapped inputs for each site, assay, and PM size fraction to allow for robust estimates of the models' terms. The lack of structure in the DFCH observations (Figure A1) resulted in poorly performing models and therefore, this assay was not included in further analyses.

When the explanatory multiple linear regression models were exposed to the PMF-identified sources it was clear that the anthropogenic road traffic and wood combustion sources had the greatest intrinsic $OP_m$ (Figure 4). When combining the five sites' results together, the road traffic and wood combustion sources were always the highest ranked $OP_m$ sources, with the exception of DTT for $PM_{2.5}$ where wood combustion was ranked first, but road traffic fell to fourth place and the nitrate-rich source was placed second. The metal-sensitive AA assay showed that the coarse-mode road traffic source was the most

potent PM source in Switzerland giving additional evidence that coarse, non-exhaust emissions drove this assay's $OP_m$ results. The mostly fine and carbonaceous wood combustion source was always important for the two $OP_m$ assays and was clearly the most potent source for $PM_{2.5}$. The other remaining six sources had, on average, positive contributions to $OP_m$, but were far less important for $OP_m$ when compared to the road traffic and wood combustion sources based on this analysis. Notably, the nitrate- and sulfate-rich sources generally showed low levels of $OP_m$ which outlines a disconnect between average PM

mass concentrations and $OP_m$ potency. This suggests that all PM has the ability to contribute to OP, but road traffic and wood combustion source are the two sources which should be prioritised for control and management to efficiently reduce $OP_v$ in Switzerland. Unlike Samake et al. (2017); Weber et al. (2021), these results do not suggest that biogenic-sourced PM is



particularly important for $OP_m$ in Switzerland, perhaps due to different fungal and plant species found in different environments or the differing intensities of agriculture and cultivation between the two countries (Samaké et al., 2019; Samake et al., 2017).

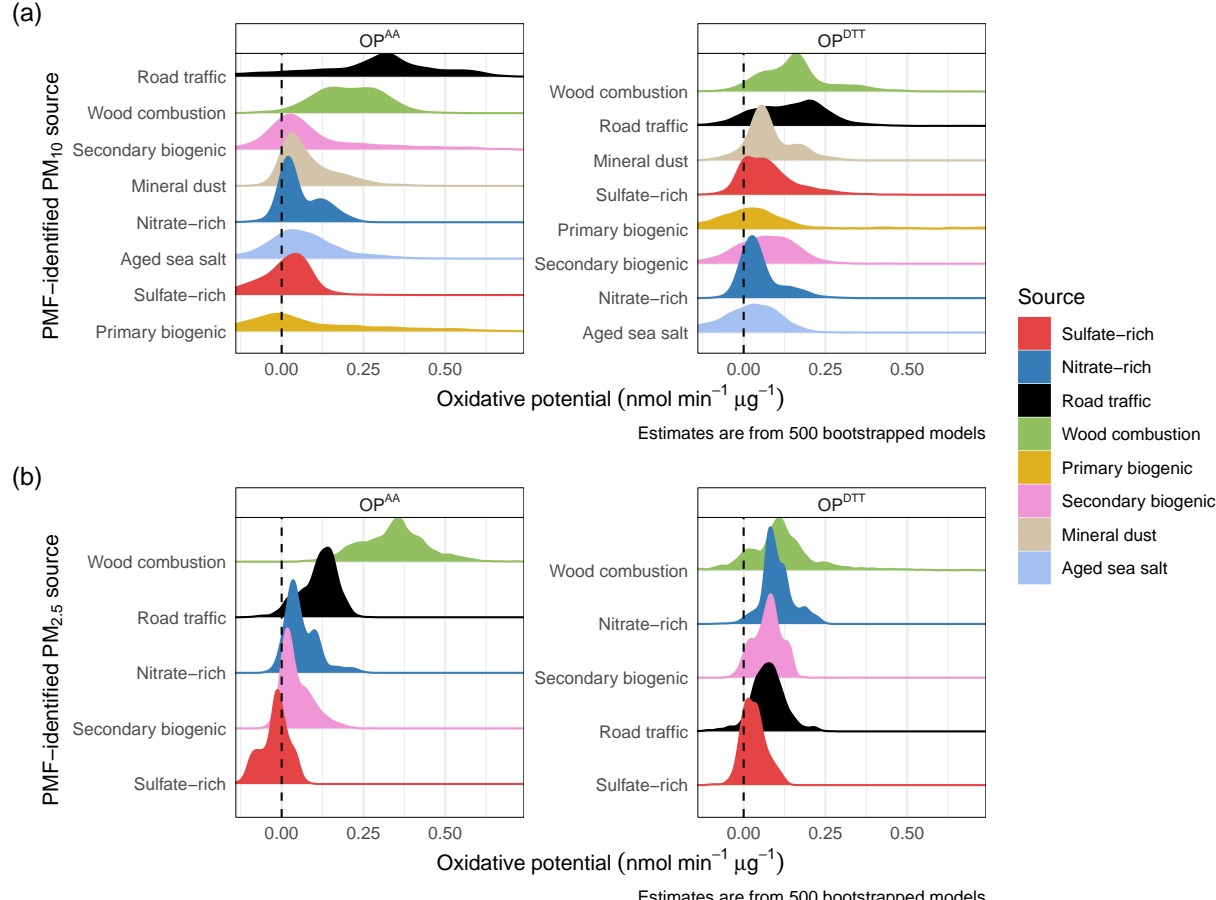

**Figure 4.** Densities of the intrinsic $OP_m^{AA}$ (a) and $OP_m^{DTT}$ (b) estimates for the eight $PM_{10}$ and five $PM_{2.5}$ PMF-identified sources. The estimates for all five sites included in the analysis have been aggregated.

## 3.3 Identifying important PM constituents with random forest

Within the PMF-identified sources shown in Figure 4, there are a large number of constituents which give the sources their characteristics. To better identify the specific components which compose the PM sources identified in Switzerland that were important and potent drivers of $OP_v$, presumably, mostly contained within the road traffic and wood combustion sources, a multi-step modelling process was conducted. The random forest algorithm was used to calculate importance and were ranked for all variables included in the data set (Breiman, 2001). A high importance ranking indicates that the variable is more important for the dependent variable's explanation and the utilisation of the random forest algorithm for this sort of application





has gained traction in many fields (Behnamian et al., 2019). The motivation for this process was to simplify and resolve the lower-level linkages between PM components and OP$_v$ when the PMF sources were potentially heterogeneous among the sampling sites and PM size fractions.

When the random forest importances were calculated for each site, PM fraction, and OP$_v$ assay, it was clear that a collection of organics and metals were commonly identified as being the most important variables for the explanation of OP$_v$. Elements and organic compounds associated with wood combustion: rubidium, potassium, levoglucosan, mannosan and galactosan were constantly ranked highly in terms of importance (Figure 5). The other group of components which were identified were metals such as copper, zinc, iron, tin, antimony, and to some extent manganese and cadmium. These collection of metals are usually as-

sociated with vehicular non-exhaust emissions and are generated by abrasive or wear processes (Charron et al., 2019; Harrison et al., 2021). EC and OC were also commonly identified and these variables are associated with both wood burning and vehicle exhaust emissions. Despite these two groups of PM constituents being identified, both mass and ions (especially nitrate) were also present in the most important variables identified by random forest. We interpret the presence of these variables as proxies of total PM mass indicating that although for a given PM mass, OP$_v$ may vary depending on its make-up, total PM mass is still

an important, and related metric. Therefore, the importance analysis was consistent with the PMF inversion process discussed in Section 3.2.

Figure 5 also shows some site specific variation due to the sites' different local emissions. For example, in Bern-Bollwerk, the non-exhaust sourced metals such as copper, iron, and zinc were ranked higher than the mean importance rank across the five sites. This feature was present in both assays and was somewhat clearer in PM$_{10}$ due to the tendency of abrasive processes

to emit PM larger than 2.5 μm (Harrison et al., 2021). Magadino-Cadenazzo on the other hand, demonstrated a tendency of rubidium, potassium, mannosan, and levoglucosan to be more important than the sites' mean ranking which was consistent with what is known about this site's exposure to local emissions because it experiences a heavy wood smoke load (Sandradewi et al., 2008; Chen et al., 2021). When comparing the two PM size fractions, there was no clear dominating source and the differences between PM$_{10}$ and PM$_{2.5}$ were overshadowed by site specific differences. This supports the conclusions made in a

companion paper (Grange et al., 2021) where non-exhaust PM$_{2.5}$ emissions were found to be considerable and are important to consider across the Swiss sampling sites. When exposing the PMF sources (eight for PM$_{10}$ and five for PM$_{2.5}$) to the same random forest importance analysis, the road traffic and wood combustion sources were clearly the most important sources for OP explanation, as shown in Figure 4.

A slightly different representation of the random forest importance rankings are provided in Figure 6, where the presence

of variables in the group which was considered highly important were counted for the five sites, two OP$_v$ assays, and two PM fractions. It is noticeable that rubidium and copper, two tracers for wood burning and non-exhaust emissions, were ranked as the most important variables for PM$_{10}$ at all five sites and for both OP$_v$ assays. For PM$_{2.5}$, where concentrations of many metals were lower than in PM$_{10}$, only a wood burning tracer (either potassium or rubidium) together with PM mass were identified across all five sites and both OP$_v$ assays. All variables which were identified more than once for each OP$_v$ assay and PM size

fraction (the variables shown in the $y$-axes of Figure 6) were used in the next step of linear modelling to identify what variables are best to be used when forming predictive models to explain OP$_v$.



**Figure 5.** Random forest importance plot of the top independent variables for two $OP_v$ assays, two particulate size fractions, and five sampling sites. The large open diamonds represent the variables' medians and the variables are ordered by their median ranking.



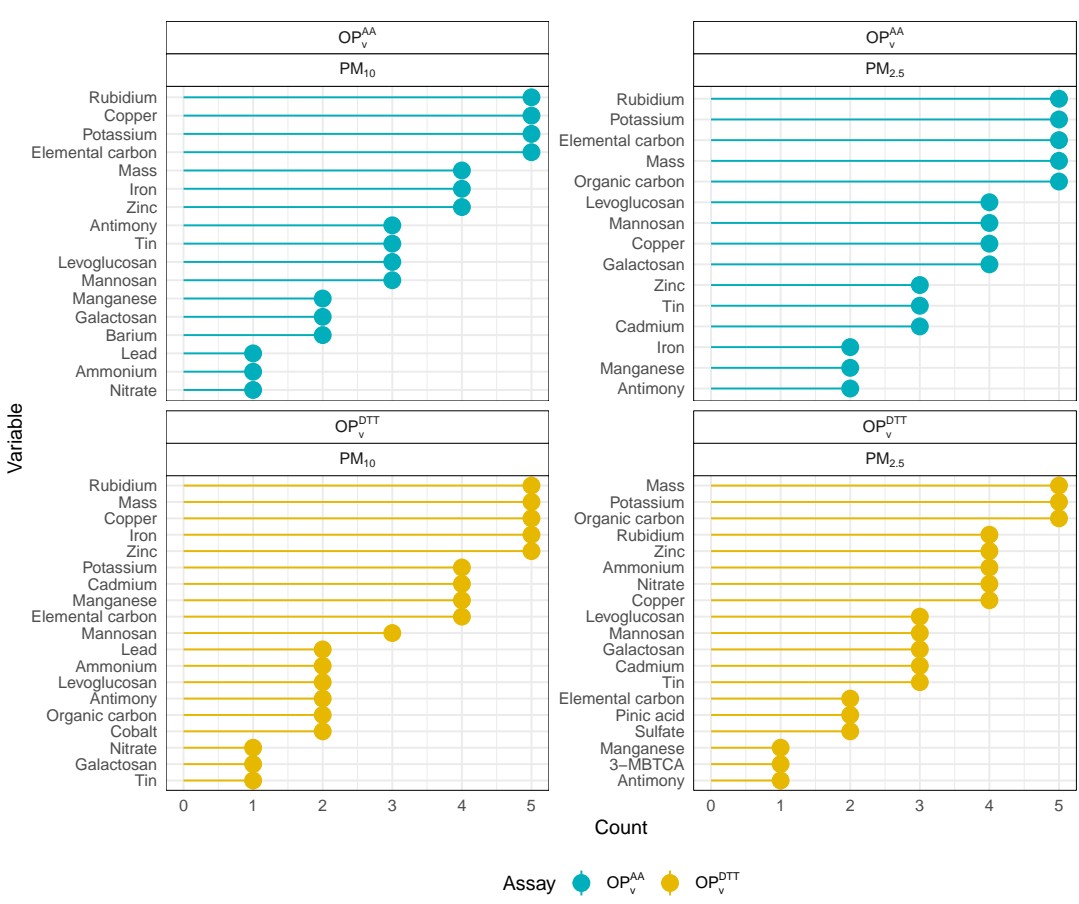

**Figure 6.** Counts of how many times an independent variable was ranked highly in terms of random forest importance for two $OP_v$ assays, two particulate size fractions, and five sampling sites. Variables with counts of five shows that for every site included in the analysis, this variable was identified as important for the explanation of $OP_v$.





### 3.4 Modelling OP

The most important variables at each site, identified by the rank of the random forest importance (Figure 6), were used to build multiple linear regression models to explain $OP_v$. Every combination of the variables were used to calculate linear regression models (with a maximum of five independent variables and the intercept terms omitted) and after training, only the models with positive slope estimates, those which had a maximum pairwise variance inflation factor (VIF) of less-than 2.5, and had an $R^2$ greater-than 75 % were kept. These three filters ensured the models selected did not suffer from undesirable levels of multicollinearity among their independent variables (Jackson et al., 2009; Barmpadimos et al., 2011) and performed adequately on their training set. The VIF filter removed all models with more than four independent variables due to the increased probability of multicollinearity when including additional independent variables in the same model. A total of 100 342 models were trained and 371 models passed the filters. The number of models trained for each site, PM fraction, and OP assay are shown in Table A2.

When analysing the models with the best performance based on their $R^2$ values, 77 % had two independent variables while models with one or three independent variables only composed 13 % and 10 % of the total set. Almost without exception, the best models' independent variables included a metal and an organic compound. The metals contained in the models were the same as those identified and discussed previously (Figure 5; Figure 6) and are generally emitted from abrasive processes related to road vehicles (iron, zinc, copper, antimony, but also cadmium), while the organics were the specific biomass burning markers of levoglucosan, mannosan, and galactosan. Table 3 shows equations of the best performing models based on their $R^2$ values for each sampling site, the two PM size fractions and the two $OP_v$. However, all models fulfilling the applied filter criteria can be considered as appropriate and considered as suitable models for explaining the observed $OP_v$. The full list of these suitable models are provided in the Supplementary Information (Table S1), the counts of all pairwise combinations of variables in the suitable models with two or more independent variables are shown in Figure 7.

The best performing models demonstrated that the *combination* of vehicular non-exhaust emission and wood burning tracers were required to generate the best models to explain $OP_v$. Interestingly, the exact tracers or markers used for the modelling was not critical. For example, using antimony, copper, or iron as the representative non-exhaust emission species resulted in models which performed very similarly and showed that these three metals were effectively interchangeable with one-another. Cadmium, manganese, and zinc could also be added to this group, but the use of these metals resulted in models which performed slight worse on average and such patterns may be related to the differing elements' analytical detection limits or the multiple emission sources these metals have. The same phenomenon was present for the wood burning tracers of levoglucosan, mannosan, and galactosan where the selection of one of these organics over the other was not critical for the explanation of $OP_v$.

Figure 7 shows, that the combinations of independent variables in the suitable models for explaining $OP_v^{AA}$ are different from those for explaining $OP_v^{DTT}$. There is clearly a larger number of combinations of independent variables in the models for $OP_v^{AA}$ compared to $OP_v^{DTT}$. The combinations of selected variables in models for $OP_v^{AA}$ are for both PM size fractions predominantly the above mentioned pairs of tracers for vehicular non-exhaust and wood burning emissions. It is interesting





**Table 3.** The best performing robust multiple linear regression model equations for each site, two PM fractions, and two OP$_v$ assays. The units used for the independent variables are $\mu g \, m^{-3}$.

| PM | Site | OP assay | $R^2$ (%) | Equation |
|---|---|---|---|---|
| PM$_{10}$ | Payerne | OP$_v^{AA}$ | 87 | OP$_v = 106.52$(galactosan) $+ 2.78$(iron) |
| PM$_{10}$ | Magadino-Cadenazzo | OP$_v^{AA}$ | 95 | OP$_v = 1.93$(levoglucosan) $+ 2.6$(iron) |
| PM$_{10}$ | Basel-Binningen | OP$_v^{AA}$ | 96 | OP$_v = 95.89$(galactosan) $+ 130.38$(copper) |
| PM$_{10}$ | Zürich-Kaserne | OP$_v^{AA}$ | 91 | OP$_v = 41.72$(mannosan) $+ 750.09$(antimony) |
| PM$_{10}$ | Bern-Bollwerk | OP$_v^{AA}$ | 89 | OP$_v = 109.01$(galactosan) $+ 318.68$(manganese) |
| PM$_{10}$ | Payerne | OP$_v^{DTT}$ | 86 | OP$_v = 151.44$(manganese) $+ 0.32$(ammonium) |
| PM$_{10}$ | Magadino-Cadenazzo | OP$_v^{DTT}$ | 87 | OP$_v = 11.81$(mannosan) $+ 134.34$(manganese) |
| PM$_{10}$ | Basel-Binningen | OP$_v^{DTT}$ | 90 | OP$_v = 1.53$(iron) $+ 0.44$(ammonium) |
| PM$_{10}$ | Zürich-Kaserne | OP$_v^{DTT}$ | 79 | OP$_v = 3.12$(potassium) $+ 0.24$(OC) |
| PM$_{10}$ | Bern-Bollwerk | OP$_v^{DTT}$ | 80 | OP$_v = 2$(EC) $+ 0.85$(ammonium) |
| PM$_{2.5}$ | Payerne | OP$_v^{AA}$ | 90 | OP$_v = 2.51$(levoglucosan) $+ 88.29$(copper) |
| PM$_{2.5}$ | Magadino-Cadenazzo | OP$_v^{AA}$ | 97 | OP$_v = 23.04$(mannosan) $+ 3.98$(iron) |
| PM$_{2.5}$ | Basel-Binningen | OP$_v^{AA}$ | 88 | OP$_v = 2.53$(levoglucosan) $+ 613.81$(antimony) |
| PM$_{2.5}$ | Zürich-Kaserne | OP$_v^{AA}$ | 91 | OP$_v = 2.91$(levoglucosan) $+ 107.98$(copper) |
| PM$_{2.5}$ | Bern-Bollwerk | OP$_v^{AA}$ | 90 | OP$_v = 17.73$(mannosan) $+ 107.21$(copper) |
| PM$_{2.5}$ | Payerne | OP$_v^{DTT}$ | 93 | OP$_v = 0.19$(organic_carbon) $+ 0.1$(nitrate) |
| PM$_{2.5}$ | Magadino-Cadenazzo | OP$_v^{DTT}$ | 85 | OP$_v = 8.25$(galactosan) $+ 0.06$(mass) |
| PM$_{2.5}$ | Basel-Binningen | OP$_v^{DTT}$ | 89 | OP$_v = 17.9$(galactosan) $+ 129.44$(copper) $+ 0.08$(nitrate) |
| PM$_{2.5}$ | Zürich-Kaserne | OP$_v^{DTT}$ | 84 | OP$_v = 2.45$(potassium) $+ 0.19$(ammonium) $+ 70.49$(pinic acid) |
| PM$_{2.5}$ | Bern-Bollwerk | OP$_v^{DTT}$ | 88 | OP$_v = 30.91$(galactosan) $+ 474.89$(tin) $+ 0.34$(ammonium) |

to note that although rubidium and potassium had higher ranks in the random forest importance, the suitable models for explaining mostly included an organic tracer for wood burning emissions (levoglucosan, mannosan, or galactosan). This could be explained by rubidium and potassium having multiple emission sources and therefore were removed by the multicollinearity filter used for the model selection.

We interpret the presence of levoglucosan, mannosan, and galactosan in this analysis as simply indicators of biomass burning emission sources. This is because these particular organic compounds are not redox-active and therefore, they cannot be the components of PM which drove OP. Quinones, rubidium, and/or other co-emitted products from biomass burning are most likely the responsible components, and this is a clear example of how an observational study can suggest and highlight associations or correlations, but not necessarily causality.

In contrast to OP$_v^{AA}$, PM mass or ammonium and nitrate were present in the better performing models for OP$_v^{DTT}$ at times. It is unlikely that ammonium and nitrate are indeed strong drivers of OP$_v$ since ammonium sulfate and nitrate ($(NH_4)_2SO_4$ and $NH_4NO_3$) and have been shown to have negligible OP (Daellenbach et al., 2020), the presence of these inorganic ions might





**Figure 7.** Number of times when a combination of two independent variables for the filtered models were present for PM$_{10}$ and PM$_{2.5}$ and two OP$_v$ assays for five sampling sites.

be acting as a proxy for total ambient PM concentrations or perhaps seasonal emission cycles due to its shift between gas and aerosol phases in the different seasons because of changes in ambient air temperature. For PM$_{2.5}$'s OP$_v^{DTT}$, OC as well as

pinic acid (a tracer for biogenic secondary organic aerosol) were frequently found in the 371 models which passed the model selection criteria. OC and pinic acid might also be understood as proxies for total PM concentrations or specific conditions



leading to elevated PM levels. Such mentioned proxies were in the models for explaining $OP_v^{DTT}$ mostly combined with an organic wood burning emission tracer and for $PM_{2.5}$ also with copper and tin.

The combinations of pairs of independent variables in suitable models for explaining $OP_v$ in $PM_{10}$ and $PM_{2.5}$ as shown in
Figure 7 indicates that the $OP_v^{AA}$ assay provided a response that was more specific to the chemical composition of PM than the $OP_v^{DTT}$ assay. It is also noticeable that for both $OP_v$ assays there are more pairwise combinations of independent variables in the suitable models for $PM_{2.5}$ than for $PM_{10}$. The reason for this observation is currently unclear and further research will be required to fully elucidate these features.

## 4   Conclusions

An intensive PM and OP sampling campaign conducted across Switzerland between 2018 and 2019 demonstrated that $OP_v$ was variable in time and space. $OP_v$ patterns followed the familiar pattern of atmospheric pollutants where urban locations were more polluted than their rural counterparts and wintertime saw enhanced $OP_v$. Although the differences between rural and urban locations were important for mass, the OP metrics constantly showed a greater difference indicating OP was more heterogeneous than PM mass across Switzerland. When comparing Switzerland's $OP_v$ with 14 sites in France where data exists and were
produced by the same sampling and laboratory procedures, Switzerland's $OP_v$ was comparable to that observed in France, but Bern-Bollwerk, a semi-canyonised urban-traffic sampling location had the highest mean $OP_v^{AA}$ (4.1 $\mathrm{nmol\,min^{-1}\,m^{-3}}$) contained in the dataset. The lack of current standardisation for OP measurement, quantification, and calibration is an issue which the air quality community should address and would allow for reliable comparisons among different locations and times in the future. The AA and DTT assays showed much more structure than the third DCFH assay which made the former approaches
more useful for data analysis than the latter.

An analysis of Switzerland's $PM_{10}$ and $PM_{2.5}$ sources identified by PMF models suggested that two major anthropogenic emission sources, namely road traffic and wood combustion were the most important drivers of $OP_v$ in Switzerland. Contrasting this was the inorganic nitrate- and sulfate-rich sources which generally had low levels of intrinsic $OP_m$ across Switzerland, as did the two biogenic sources (primary and secondary). This outlines the potential disconnect between total PM mass concentration and $OP_m$ which has been noted by others, for example, Daellenbach et al. (2020) and this observation may update
the management priorities of PM sources with a focus on health impacts rather than total mass.

Further investigation into the components of PM using a random forest dimensionality reduction technique and multiple linear regression models demonstrated that a collection of metals associated with non-exhaust emissions such as copper, zinc, antimony, iron, tin, manganese, and cadmium as well as the specific wood combustion tracers of levoglucosan, mannosan, and
galactosan (or associated elements such as rubidium and potassium) were consistently important for the explanation of $OP_v$. The combination of a non-exhaust sourced metal and a biomass burning tracer provided very good models which could explain $OP_v$ well when considering their training sets. The observations also suggested that $OP_v^{AA}$ was a more specific or sensitive OP metric than the other assays employed, in the Swiss locations where sampling took place. To consider $OP^{AA}$ as a future, standard OP metric, additional evidence of associated health outcomes will be required, however.





The results above point towards the need to control wood burning sourced PM and non-exhaust emissions to reduce the $OP_v$ of Switzerland's atmospheres. Such conclusions are not out of step with current air quality management practices and priorities, but reinforce the importance of these sources and their respective chemistry in respect to $OP_v$ – potentially a health relevant metric for PM. Therefore, a renewed focus on wood burning and non-exhaust emissions is encouraged to reduce the deleterious heath effects of PM. Because non-exhaust emissions and wood burning emissions can be effectively controlled at

a local level, it is likely that significant reductions of $OP_v$ could be achieved without the need for regional and trans-boundary management collaboration.

    The causality of the identified sources (and PM constituents) for driving $OP_v$ could always be questioned because the biological mechanisms which result in pathology were not investigated with this observational study. However, the results are consistent with those found in the literature and gives very clear suggestions on where to focus future efforts to identify the

linkage between biological mechanisms and $OP_v$. It is also clear that the $PM_{10}$ and $PM_{2.5}$ size fractions have different $OP_v$ characteristics and the $OP_v$ is not simply additive. Furthermore, considering the importance of non-exhaust emissions for the coarse-mode, the importance of continued $PM_{10}$ monitoring is outlined.





**Figure A1.** Time series of the three OP$_v$ assays for PM$_{10}$ and PM$_{2.5}$ at five sampling sites in Switzerland between June, 2018 and May, 2019.



**Table A1.** Annual means of PM$_{10}$ OP$_\mathrm{v}^\mathrm{AA}$ and OP$_\mathrm{v}^\mathrm{DTT}$ for sampling sites in France and Switzerland where identical methods to quantify OP$_\mathrm{v}$ has been conducted. The units used for the means is $\mathrm{nmol\,min^{-1}\,m^{-3}}$ and the 14 sites' data from France are from Weber (2021); Weber et al. (2021).

| Rank | Country | Urban area | Site | Site type | OP assay | Mean |
|---|---|---|---|---|---|---|
| 1 | Switzerland | Bern | Bern-Bollwerk | Traffic | OP$_\mathrm{v}^\mathrm{AA}$ | 4.1 |
| 2 | France | Chamonix | CHAM | Urban valley | OP$_\mathrm{v}^\mathrm{AA}$ | 2.3 |
| 3 | France | Nogent | NGT | Urban background | OP$_\mathrm{v}^\mathrm{AA}$ | 2.2 |
| 4 | France | Passy | PAS | Urban valley | OP$_\mathrm{v}^\mathrm{AA}$ | 2.2 |
| 5 | France | Roubaix | RBX | Traffic | OP$_\mathrm{v}^\mathrm{AA}$ | 2.1 |
| 6 | Switzerland | Zürich | Zürich-Kaserne | Background | OP$_\mathrm{v}^\mathrm{AA}$ | 1.7 |
| 7 | France | Aix-en-provence | AIX | Urban background | OP$_\mathrm{v}^\mathrm{AA}$ | 1.7 |
| 8 | Switzerland | Cadenazzo | Magadino-Cadenazzo | Background | OP$_\mathrm{v}^\mathrm{AA}$ | 1.7 |
| 9 | France | Grenoble | GRE-fr_2013 | Urban background | OP$_\mathrm{v}^\mathrm{AA}$ | 1.7 |
| 10 | France | Marnaz | MNZ | Urban valley | OP$_\mathrm{v}^\mathrm{AA}$ | 1.6 |
| 11 | France | Vif | VIF | Urban background | OP$_\mathrm{v}^\mathrm{AA}$ | 1.5 |
| 12 | France | Grenoble | GRE-fr_2017 | Urban background | OP$_\mathrm{v}^\mathrm{AA}$ | 1.5 |
| 13 | France | Grenoble | GRE-cb | Urban background | OP$_\mathrm{v}^\mathrm{AA}$ | 1.4 |
| 14 | France | Strasbourg | STG-cle | Traffic | OP$_\mathrm{v}^\mathrm{AA}$ | 1.3 |
| 15 | Switzerland | Basel | Basel-Binningen | Background | OP$_\mathrm{v}^\mathrm{AA}$ | 1.2 |
| 16 | France | Talence | TAL | Urban background | OP$_\mathrm{v}^\mathrm{AA}$ | 1.0 |
| 17 | France | Nice | NIC | Urban traffic | OP$_\mathrm{v}^\mathrm{AA}$ | 1.0 |
| 18 | Switzerland | Payerne | Payerne | Background | OP$_\mathrm{v}^\mathrm{AA}$ | 0.7 |
| 19 | France | Port-de-Bouc | PdB | Industrial | OP$_\mathrm{v}^\mathrm{AA}$ | 0.6 |
| 20 | France | Marseille | MRS-5av | Urban background | OP$_\mathrm{v}^\mathrm{AA}$ | 0.5 |
| 1 | France | Passy | PAS | Urban valley | OP$_\mathrm{v}^\mathrm{DTT}$ | 4.4 |
| 2 | Switzerland | Bern | Bern-Bollwerk | Traffic | OP$_\mathrm{v}^\mathrm{DTT}$ | 2.9 |
| 3 | France | Grenoble | GRE-fr_2013 | Urban background | OP$_\mathrm{v}^\mathrm{DTT}$ | 2.7 |
| 4 | France | Nogent | NGT | Urban background | OP$_\mathrm{v}^\mathrm{DTT}$ | 2.7 |
| 5 | France | Roubaix | RBX | Traffic | OP$_\mathrm{v}^\mathrm{DTT}$ | 2.6 |
| 6 | France | Marseille | MRS-5av | Urban background | OP$_\mathrm{v}^\mathrm{DTT}$ | 2.6 |
| 7 | France | Strasbourg | STG-cle | Traffic | OP$_\mathrm{v}^\mathrm{DTT}$ | 2.4 |
| 8 | France | Chamonix | CHAM | Urban valley | OP$_\mathrm{v}^\mathrm{DTT}$ | 2.3 |
| 9 | France | Nice | NIC | Urban traffic | OP$_\mathrm{v}^\mathrm{DTT}$ | 2.2 |
| 10 | France | Aix-en-provence | AIX | Urban background | OP$_\mathrm{v}^\mathrm{DTT}$ | 1.9 |
| 11 | France | Talence | TAL | Urban background | OP$_\mathrm{v}^\mathrm{DTT}$ | 1.8 |
| 12 | France | Marnaz | MNZ | Urban valley | OP$_\mathrm{v}^\mathrm{DTT}$ | 1.8 |
| 13 | France | Port-de-Bouc | PdB | Industrial | OP$_\mathrm{v}^\mathrm{DTT}$ | 1.8 |
| 14 | France | Grenoble | GRE-cb | Urban background | OP$_\mathrm{v}^\mathrm{DTT}$ | 1.7 |
| 15 | France | Grenoble | GRE-fr_2017 | Urban background | OP$_\mathrm{v}^\mathrm{DTT}$ | 1.5 |
| 16 | Switzerland | Zürich | Zürich-Kaserne | Background | OP$_\mathrm{v}^\mathrm{DTT}$ | 1.3 |
| 17 | France | Vif | VIF | Urban background | OP$_\mathrm{v}^\mathrm{DTT}$ | 1.3 |
| 18 | Switzerland | Cadenazzo | Magadino-Cadenazzo | Background | OP$_\mathrm{v}^\mathrm{DTT}$ | 1.0 |
| 19 | Switzerland | Payerne | Payerne | Background | OP$_\mathrm{v}^\mathrm{DTT}$ | 0.8 |
| 20 | Switzerland | Basel | Basel-Binningen | Background | OP$_\mathrm{v}^\mathrm{DTT}$ | 0.8 |





**Table A2.** The number of multiple linear regression (MLR) models trained for each site, PM size fraction, and OP assay. The total number of models was 100 342.

| Site | PM fraction | OP assay | Number of models trained |
|---|---|---|---|
| Basel-Binningen | $PM_{10}$ | $OP_{AA}$ | 2379 |
| Basel-Binningen | $PM_{10}$ | $OP_{DTT}$ | 6884 |
| Basel-Binningen | $PM_{2.5}$ | $OP_{AA}$ | 4943 |
| Basel-Binningen | $PM_{2.5}$ | $OP_{DTT}$ | 6884 |
| Bern-Bollwerk | $PM_{10}$ | $OP_{AA}$ | 3472 |
| Bern-Bollwerk | $PM_{10}$ | $OP_{DTT}$ | 6884 |
| Bern-Bollwerk | $PM_{2.5}$ | $OP_{AA}$ | 4943 |
| Bern-Bollwerk | $PM_{2.5}$ | $OP_{DTT}$ | 6884 |
| Magadino-Cadenazzo | $PM_{10}$ | $OP_{AA}$ | 2379 |
| Magadino-Cadenazzo | $PM_{10}$ | $OP_{DTT}$ | 6884 |
| Magadino-Cadenazzo | $PM_{2.5}$ | $OP_{AA}$ | 4943 |
| Magadino-Cadenazzo | $PM_{2.5}$ | $OP_{DTT}$ | 6884 |
| Payerne | $PM_{10}$ | $OP_{AA}$ | 2379 |
| Payerne | $PM_{10}$ | $OP_{DTT}$ | 4943 |
| Payerne | $PM_{2.5}$ | $OP_{AA}$ | 3472 |
| Payerne | $PM_{2.5}$ | $OP_{DTT}$ | 4943 |
| Zürich-Kaserne | $PM_{10}$ | $OP_{AA}$ | 3472 |
| Zürich-Kaserne | $PM_{10}$ | $OP_{DTT}$ | 4943 |
| Zürich-Kaserne | $PM_{2.5}$ | $OP_{AA}$ | 4943 |
| Zürich-Kaserne | $PM_{2.5}$ | $OP_{DTT}$ | 6884 |

**Table S1.** Attached `op_model_collection_which_passed_the_filters.csv` file containing all linear regression model formulations with their estimates and model statistics.



*Data availability.* The data sources used in this work are described and some data sets are publicly accessible in a persistent data repository (Grange, 2021a, https://doi.org/10.5281/zenodo.4668158). Additional data and information are available from the authors on reasonable
request.

*Author contributions.* SKG and CH conceived the research questions and wrote the manuscript. SKG conducted the data analysis and GU managed the OP laboratory analyses. GU, SW, and JJ helped with revising and improving the manuscript.

*Competing interests.* The authors declare no competing interest.

*Acknowledgements.* This work was funded by the Federal Office for the Environment (FOEN) [contract number: 16.0096.PJ/R152-0739].
The authors thank the wider *Projekte Quellenzuordnung Feinstaub* team for their contributions. SKG is also supported by the Natural Environment Research Council (NERC) while holding associate status at the University of York. Andrés Alastuey and Xavier Querol from Institute of Environmental Assessment and Water Research, Consejo Superior de Investigaciones Científicas are thanked for their help with the elemental ICP analysis. GU and JLJ thank the ANR-15-IDEX-02, ANR-19-CE34-0002-01, and Foundation of University of Grenoble Alpes for the funding of instruments on the AirOSol analytical plateau at IGE.



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
