# Peer review of "Linking Switzerland's $PM_{10}$ and $PM_{2.5}$ oxidative potential (OP) with emission sources"

_Atmospheric Chemistry and Physics, 2021_

## Author Response (AR1)

**Author responses to reviewer's comments of** `acp-2021-979` (*Linking Switzerland's PM$_{10}$ and PM$_{2.5}$ oxidative potential (OP) with emission sources*)

April 20, 2022

Stuart K. Grange[*], Gaëlle Uzu, Samuël Weber, Jean-Luc Jaffrezo, Christoph Hueglin

[*]stuart.grange@empa.ch

**Anonymous referee #1**

**General comments**

This paper uses filter samples collected at a number of sites in Switzerland over a period of roughly a year to determine particle oxidative potential by three methods for PM2.5 and PM10. The paper adds to a growing list of research that assumes OP is a relevant metric to relate aerosol chemical properties to adverse effects on human health, although they present no data on this in this paper. The paper investigates what sources contribute most to the measured OP and claim to investigate what the chemical components are that drive it (objective (ii) line 90). They also assess the performance of the assays, concluding that one of them is a more specific metric, which is interpreted, I guess, to mean a better connection to human health (more on this below)? The paper essentially reinforces the view that with declining vehicle tail pipe emissions, the two main emission sources of concern are particles emitted from vehicle brakes and abrasion of tires and roadways and wood burning (residential). The paper, however, does not really address what specific compounds drive the OP since they do not sufficiently chemically speciate the OA (they only find traces of wood burning linked to OP (i.e., sugars), which are likely only markers, and they do not actually measure metal ions which are the species that would be driving the OP assays. (For example, the lines 370-374 really apply to all the specific PM components measured, including the metals). The paper is well written and the data is interesting. However, there are areas where the authors could provide more clarity, but overall, it is an important contribution.

Many thanks for your positive comments. Below are our responses to your itemised comments.

**Itemised comments**

1. Simple linear models involving various species are used to reconstruct the observed OP levels, and they seem to do a reasonable job given the high r2. However, other research has shown that OP measured in a filter extract is not likely simply due to the sum of individual species, there can be interaction between species. Why does this appear not to be an issue in this data set?

   We agree that the composition of the PM mix will give the PM its OP characteristics. However, the modelling approach used here had the objective at identifying what components are significantly correlated with OP. No predictions are made with the models that were created and this application would likely require more investigation into the interaction among the PM's constituents to be accurate.

2. Few details are given on the assays. I think it would be useful at a minimum to note what the filter extraction liquid was for each assay (it was not water, simulated lung fluid for DTT and AA, but not DCFH?) and why was this done, e.g., Calas et al. 2017 does not appear to show a difference in ambient samples for the OP_DTT assay in pure water vs. SLF, so what is the justification? It would be useful to explain and support why this is claimed to be important given that the authors make the point many times that lack of standard methods impedes this area of research. Also, how is the DCFH assay used so that the results are reported in units similar to the other two assays (ie, does DCFH assay really have units of nmol/min/m3, last line of section 2.3.1.) — this does not seem possible? And finally, the authors state that a liquid concentration of roughly 25 ug/L was used for the OP tests, but in their methods cited (Calas et al., 20218), a concentration of 10 ug/mL was used instead, a huge difference (ug/L vs ug/mL)?

In light of this (and other) feedback, the text describing the OP assays has been expanded and edited. The text now describes the filter extraction liquid in detail.

The unit for the DCFH assay is nmol $H_2O_2$ m$^{-3}$. This is now clearly stated in Section 2.3.1 and in the figure labels and captions when needed.

The unit for extraction concentration was also incorrect. This has been corrected to 25 µg mL$^{-1}$ and justification of 25 rather than 10 µg mL$^{-1}$ is now given. These paragraphs have been substantially expanded and edited. These changes are better viewed in the accompanying 'diff' file.

In Calas et al 2017,[1] based on the limited series of PM samples, OP results were statistically different only between ALF and Milli-Q water (and not between Milli-Q water and Gamble's solutions). It was claimed that given the evidenced mechanisms in both Gamble's and ALF solutions and their potential responsibility in the significant difference for OP measured in ALF and Milli-Q water, the observed low bias for OP measured in Gamble's solutions is probably more than just random differences and could actually result in significant differences in other environments, as such differences exist among reference compounds. All the more, the decrease or increase of OP values for reference compounds after SLF extraction was unpredictable compared to water extraction, and that, this could be a significant source of biases in health outcome studies that could be addressed by further improvements in OP assays.

3. Relating to the above, what does "All extracts were conducted at iso-and low-mass PM concen-

[1]Calas, A. et al. (2017). The importance of simulated lung fluid (SLF) extractions for a more relevant evaluation of the oxidative potential of particulate matter. *Scientific Reports* 7.1, p. 11617. URL: https://doi.org/10.1038/s41598-017-11979-3.

tration...' mean. (What is the meaning of iso-...?)

Iso- in this context means the same over time. This paragraph has been substantially edited due to the comments above and now reads:

"In order to maintain a constant amount of extracted PM, filter punches were adjusted by area to obtain an iso-mass of $25\,\mathrm{\mu g\,mL^{-1}}$ to ensure intercomparison among the samples."

4. Line 375 to 380 relating to the inclusion of ammonium nitrate in the models. The reason could be more complex than what is stated, e.g., ammonium nitrate could largely control particle water, which in turn controls aqueous reactions that affect OP, or ammonium nitrate could just be a tracer for secondary processes in general or more photochemically aged aerosol, which has been shown to affect metals solubility and OP (eg, Wong et al, Environ. Sci. Technol. 2020, 54, 7088–7096; Antinolo, et al.,2015, Nature Comm, 6(6812 ), 1-7; Li et al, 2013 Atmos. Env., 81, 68-75; Zhu et al, 2020, Envir. Sci Technol., 54, 8558-8567, and others on formation of secondary OP species) The point is, atmospheric aging alters aerosol particle chemical properties. When interpreting their data, the view by the authors seems to be that almost everything as primary.

This is an important point and was indeed somewhat neglected in the original manuscript. To address this, text has been edited thought the manuscript and an additional paragraph in the discussion has been added. This additional paragraph uses the references supplied and outlines why this is an important process to understand and the limitations of this current work to address these secondary processes.

"An alternative or supplementary interpretation of the above observations is that atmospheric ageing of PM and the changes that such processes induce modify the OP character of PM. Indeed, the importance of secondary PM ageing for OP has been shown by other work (Antiñolo et al., 2015; Wong et al., 2019; Zhu et al., 2020). Future studies will need to be conducted to further fully understand these processes, however. This analysis is limited by the PM sampling campaign and associated PMF models that were able to be produced in this analysis. Further understanding of secondary PM sources and OP would be very useful to fully understand OP dynamics across Switzerland."

5. Finally, in the Abstract and at the end of the paper it is stated that AA may be a more specific metric for OP than the other assays. This is apparently based on the idea discussed starting on line 362; that the linear models for predicting OP_AA at the various measurements sites have a wider range of tracers in them, but they still all point to the same source, brakes/road dust and wood burning. What does that mean; that AA really is only influenced be species from these two sources, whereas the other assays are also sensitive to other species that may not be from these sources? One could interpret this as; the AA assay is sensitive to fewer sources for OP (not a good thing), or that it does not include influences from species that have no effect

on OP (that would be a good thing)? How do the authors even know how to decide this so as to determine what is the better assay, there is no evidence shown that these are the only main sources that produce adverse health effects under the oxidative stress paradigm (there is no health data presented in this paper)? One may argue the opposite, that actually an OP assay that is more comprehensive, that includes more sources that can contribute to oxidative stress, is ideal. Note, DTT included NH4HO3, but AA did not, and see comments above on this. The discussion (or argument) on the relative merits of these three assays based on the findings of this study should be discussed more thoroughly. The current logic for the assessment of these assays is not clear to me.

After careful consideration of this comment, we agree that more evidence is required for our original statements. The alternative interpretation by the reviewer is also valuable and will require investigation of health endpoints. Therefore, a sentence in the abstract and two sentences in the conclusions have been removed.

**Anonymous referee #2**

**General comments**

Grange et al. investigated the oxidative potential (OP) of ambient PM10 and PM2.5 at five sites located in different environments of Switzerland. OP was assessed using three different endpoints - ascorbic acid (AA) and dithiothreitol (DTT) consumption, and dichlorofluorescein (DCFH) assay. They explored the spatiotemporal variability of OP and compared the OP levels with those measured in France. The source analysis followed by the investigation of sensitive components were then conducted among measured PM species, mass, and OP, and the finding suggests a higher level of OPm associated with non-exhaust traffic emissions and wood burning.

Overall, this paper provides a thorough analysis of OP levels in Switzerland and its comparison with France. The selection of sites covered common types of geographical locations with high population, and the comparison of OP with those in France provided a wider spatial scope of the health effects of PM10 in Europe. The results of sources with high OPm potency identified in this paper were generally consistent with previous studies in different regions, and the results pointed towards the importance of controlling local traffic emissions and woodburning. However, the protocol of sensitive chemical identification seemed to be flawed. By filtering the species with multicollinearity, some important contributing species might be also filtered. The regressions with significant R2 did not present the important contributors, which might indicate the lack of scientific implications of the results from this method. Furthermore, the species that were found to be highly correlated with OP in the multiple linear regression were not critical and could be easily replaced by other species, further pointed out the weakness of this method. Therefore, I would like to recommend that the paper should be majorly revised majorly for further consideration.

Many thanks for the feedback and the comments. They are very useful for enhancing the description of the applied methods and the discussion of the results for improving the readability of the manuscript. However, we do not agree with all of the reviewer's comments and conclusions. Please see below our point to point responses to the comments and our suggestions for the revision of the manuscript.

**Itemised comments**

**Major comments**

1. The materials and methods section written in the paper were too concise. The characteristics of five sampling sites, the list of measured chemical species on filters, selection criteria applied in random forest, and the factors used in PMF are not provided, and should be further described to provide a full understanding of all the protocols.

We have revised the method's section and expanded the text to include further details. We believe that much of what the reviewer desires is contained in a reference (Grange et al. (2021)) that was unable to be accessed at the time of review (see point and response below). Therefore, the text more clearly points towards Grange et al. (2021) for further details on the characteristics of the sampling sites and the full list of measured chemical species. Discussion on the PMF approach has been expanded to include the identified factors/sources, however. Please see the point below that discusses the random forest importance method more fully.

2. The level of PM used in this study (25 ug/L, i.e. 0.025 ug/mL) is far lower than that applied in most other studies (10-50 ug/mL). I highly doubt that this level may generate valid OP results. Please check the unit of the concentrations.

   The unit for extraction concentration was indeed incorrect. This has been corrected to $25\,\mu g\,mL^{-1}$.

3. While explaining that AA is a major constituent of lung lining fluid, the paper used an AA-only model for monitoring the consumption of AA, but many studies used the endpoint of AA in surrogate lung lining fluid to better represent the biological environment in human. Please provide a justification of using the AA-only model.

   In other studies, considering RTLF assay (we assume this is what the referee highlighted), the mix of AA and gluthatione, both most common lung anti-oxidants is presented as surrogate lining fluid itself (which is more simplistic and less complete than our approach). AA and gluthatione are usually measured after $4\,h$ of reaction (endpoint), because their kinetics can not be approached (competitive aspects, preferential reaction and wavelength incompatibility to follow both kinetic depletions).

4. Figure 2 combining all three endpoints in the same box seems to be confusing. Since the comparison among these endpoints is not reasonable, I would suggest splitting them into different boxes as per different endpoints, for presenting the data.

   Figure 2 has been replotted with different panels for each site and each OP assay to enable consistent comparisons among the three different OP assays.

5. Grange et al. 2021 is not available online and it seems to contain a lot of information for the interpretation of this paper.

   We apologise that this reference was not available during the initial review. This paper has been accepted during the early review process and can now be publicly accessed.[2] The bibliography has been updated to reflect this change in the reference.
* * *
[2]Grange, S. K. et al. (2021). Switzerland's $PM_{10}$ and $PM_{2.5}$ environmental increments show the importance of non-exhaust emissions. *Atmospheric Environment: X* 12, p. 100145. URL: https://www.sciencedirect.com/science/article/pii/S2590162121000459.

6. In lines 218-221, the authors explained the trend of OPcoarse, but this term is not well defined or calculated anywhere in the entire manuscript. Therefore, I suggest to provide further description, trend and calculation of this term.

$PM_{coarse}$ was not introduced in the original manuscript. As requested by Referee #3's comment number 5, we now also provide more information on the share of $PM_{coarse}$ in total $PM_{10}$ and the contribution of OP in $PM_{coarse}$ to the total OP signal in $PM_{10}$. The text now reads:
"Another key observation from these aggregations was that $PM_{coarse}$, defined as the mass concentration of PM with a size between 2.5 and 10 $\mu$m, contained 50 and 45 % of the $OP_v^{AA}$ and $OP_v^{DTT}$ signal respectively. This was only able to be highlighted because the sampling design included both $PM_{10}$ and $PM_{2.5}$."

7. The discussion of PMF results lacks depth. Even if most of the results might have been provided in Grange et al. (2021) which is not accessible now, the discussion for the MLR analysis between OP and PM sources should be enhanced. The contribution of sources towards different OP endpoints in OPv should be involved in this section.

As stated above, it was unfortunate and we apologise for the lack of access to Grange et al. (2021), where the PMF results are presented and discussed in detail. We believe that the brief summary of the PMF results is appropriate and should not be elaborated on in this paper. An addition has been made in the revised text where the relative contributions of the secondary biogenic source is now given. For description and discussion of the applied MLR we agree with the referee, more details on both the methodology and also the model performance (sections 2.5. and 3.2) are now given.

8. The method of random forest is very ambiguous: the selection of importance based on ranking should be provided in the paper. Also, in Figure 6, the justifying criteria "ranking highly" should be quantified.

The two paragraphs that explain the random forest procedure and variable importances in the methods section has been evaluated and edited to add more information. The logic regarding the section of variables based on the importance metric was already present, but the text has been expanded to make the section criterion clearer.
The text and Figure 6's caption has been edited to define ranked highly as the top 12 variables.

9. The results showing interchangeable species for the significant correlations between PM OP and concentration of components is concerning: the actual contributing species might be omitted during the selection and the final results could only find out the indicators towards important sources. Although some key contributing species like Cu and Mn were identified, they were eclipsed in the numerous correlation pairs of non-contributing sources and OP. This is

further demonstrated by Figure 7: the pairs of species involving most significant correlations (Sb and galactosan) were not contributing to PM OP. Therefore, the causality is not indicated by this method. This should be included in the discussion of the limitations.

We disagree with the reviewer's conclusion around the interchangeable nature of the species is concerning. It is clear and well known that regression analysis allows the investigation of correlations between two quantities, here PM OP and chemical PM constituents, but not the proof of causality. It is not possible and not the goal of this work to investigate causality, one of the main objectives instead is the identification of chemical constituents that are highly correlated with PM OP. Interchangeability of species must, however, be expected, as the concentrations of those chemical species that are predominantly stemming from the same emission source(s) are highly correlated. It is the observed interchangeability in the regression models that indicate the type of sources that are likely driving the OP of PM. Therefore, we argue that the interchangeability of the marker-species is a strength of the analysis because it reinforces the conclusions of the PMF-identified sources. We are careful not to overinterpret the results and state clearly that causality is not assumed in this analysis. The text reads:
"...and this is a clear example of how an observational study can suggest and highlight associations or correlations, but not necessarily causality."

10. Including PM mass in the regression might not be a good idea: PM mass might have a much higher weight than the chemical species included in the model. Therefore, the results might be biased, since OPv is determined by PM2.5 mass to some extent. Therefore, I would suggest removing PM mass for the MLR analysis between OP and species.

We also disagree with the reviewer on this point. Maybe it was not clear in the text that all possible models using all combinations of independent variables have been tested. Thus, over one-hundred thousand models were run with a range of independent variables and some models included PM mass while others did not. The presence of mass in Figure 5 and Figure 6 indicates that PM mass was not totally decoupled from OP, which is an important observation. The inclusion of mass in the modelling process does not introduce a bias in the results.

**Minor comments**

1. The introduction section should be supported by more literature. For example, in line 66-67, the statement of the different spatiotemporal trends of OP and PM mass could be supported by Yang et al. (2015) (DOI: 10.1016/j.atmosenv.2014.11.053), Liu et al. (2018) (DOI: 10.1016/j.envpol.2018.01.116) and Yu et al. (2021) (DOI: 10.5194/acp-21-16363-2021).

Done. These three references have been added to this paragraph.

2. Line 37: remove "say,"

3. Please provide further details of DCFH assay, including the cells used for this assay and assay protocols.

This section has been reviewed and the assay's protocols are present:

"The 2,7-dichlorofluorescin (DCFH) assay is commonly used for detecting intracellular $H_2O_2$ and oxidative stress using a non-fluorescent probe through the formation of a fluorescent product (dichlorofluorescein or DCF) in the presence of ROS and horseradish peroxidase (HRP). DCF was measured by fluorescence at the excitation and emission wavelengths of 485 and 530 nm, respectively, every 2 min for a total of 30 min of analysis time. The ROS concentration in the sample is calculated in terms of $H_2O_2$ equivalent based on a $H_2O_2$ calibration (100, 200, 300, 400, 500, 1000, and 2000 nmol)."

4. Line 148: Please provide the supporting citations for the statement 'DCFH assay is sensitive to organic compounds".

This text has been edited to reflect DCFH's sensitivity to compounds associated with secondary aerosol. Many of the compounds are organics, but the references that have been used are more general and simply state "secondary aerosol" rather than secondary organic aerosol. The text now reads:

"DCFH shows a preferential sensitivity to a number of compounds that are often associated with secondary aerosol (Perrone et al., 2016; Pietrogrande et al., 2019)."

5. Line 158: Provide the full name of "SOURCES". Also, this sentence is highly confusing: is PMF informatively known as extended PMF, or is SOURCES program informatively known as extended PMF? Think restructuring this sentence.

This sentence has been restructured to enhance clarity. The SOURCES project is not an acronym and is simply the name of the project that is referenced. The sentence now reads:

"The PMF approach employed is informally known as "extended PMF" and was a result of the SOURCES research programme that involved the development of harmonised PMF methodology across several sites in France (Favez et al., 2017; Weber et al., 2019, 2021)."

6. Line 220: Saying that OPcoarse is biological relevant is not rigorous since the biological relevance should not only consider the level of OP but also include the accessibility of these coarse particles in the respiratory tract. Suggest revising this statement.

True, we have edited this sentence and it now reads:

"This point implies that $PM_{coarse}$ is potentially relevant for human health and for regulatory purposes. Therefore, it is important to continue $PM_{10}$ monitoring in addition to the measurement of $PM_{2.5}$."

7. Line 237: The comparison did not involve OP DCFH This information should be listed.

   DCFH observations are unavailable for the French sites discussed, therefore such comparisons are unable to be done currently. A sentence has been added stating this:
   "Additionally, comparisons of the DCFH assay were unable to be conducted due to a lack of available data for the French sites."

8. Line 252: This sentence should be moved to the discussion of OPAA in the previous paragraph.

   This sentence was evaluated and it was decided it was duplicated information, and therefore has been removed.

9. Line 386: Provide the comparison of numbers of pairwise combinations between PM2.5 and PM10.

   The text has been edited and now includes these counts.
   "It is also noticeable that for both $OP_v$ assays there are more pairwise combinations of independent variables in the suitable models for $PM_{2.5}$ than for $PM_{10}$ (for $OP_v^{AA}$: 74 vs. 111 and for $OP_v^{DTT}$: 74 vs. 106)."

**Anonymous referee #3**

**General comments**

Grange et al. presented three OP measurements at five different sites in Switzerland. Spatial and seasonal variations are discussed. A MLR model was then used to link OP to PM sources resolved with PMF and the results suggest that road traffic and wood combustion are the major sources. Lastly, the authors used a machine learning algorithm to identify the most important PM constituents to explain OP-DTT and OP-AA to be non-exhaust metals and wood burning organic tracers. Overall, the work is interesting while more details and explanations are needed. I recommend acceptance with some minor revisions.

Thank you for your positive comments. Please see the itemised points below for our responses to your specific pieces of feedback.

**Itemised comments**

1. line 129, the consumption of AA could also be due to direct reactions between PM components with AA for example metals.

   This sentence has been edited to include this process.

   "The AA assay relies on one of the main lung antioxidants, ascorbic acid. The consumption of AA ($nmol\,min^{-1}$) in the assay is inferred as the OP of PM quantified by the transfer of electrons from AA to oxygen, or the direct reaction between PM components and AA."

2. section 2.5, it is still not clear how OP are linked to PM sources identified from PMF using MLR. More details are needed. What are the results for slope coefficients? Are the model interpretation based on the assumption that the sources contribute linearly to OP? How was OPm calculated? OP was not included in the PMF models. Won't it be more reasonable to just include OP in the PMF model? It would also be helpful to see if including OP in the model result in a better representation of OP-DCFH.

   The methods section (Section 2.5) has been expanded to include a reference to the method. When these results are discussed in Section 3.2, extra information about the models' $R^2$ and residuals are also mentioned.

   This procedure does assume linear relationships among the PMF-identified sources and OP. Admittedly, this assumption may not ideal and has been mentioned elsewhere.[3]

   $OP_m$ was calculated as part of the laboratory analysis because the mass of the PM that is sampled was known. $OP_v$ was calculated from $OP_m$ by normalising the measurement by the air
* * *
[3]Weber, S. et al. (2018). An apportionment method for the oxidative potential of atmospheric particulate matter sources: application to a one-year study in Chamonix, France. *Atmospheric Chemistry and Physics* 18.13, pp. 9617–9629. URL: https://www.atmos-chem-phys.net/18/9617/2018/.

volume that passed through the filter during the PM sampling.

An alternative approach would be to include the OP variables into the PMF modelling process. However, this was not done because it would, by definition, ensure a positive contribution of OP, would potentially alter the PMF solution, and would make the approach less flexible in terms of other uses of the PMF models and inversion methods. This too has been discussed elsewhere.[4]

We agree that there is an outstanding issue with the $OP^{DCFH}$ assay. However, further work will be required on this metric and will most likely require a different analysis approach to fully leverage these observations.

3. Rubidium seems to rank top 4 in all cases. This has never been found in any other previous studies, to my best knowledge. Is Rubidium DTT- and AA-active or is it linked to OP sources? What are the sources of Rubidium in Switzerland? Please provide references that indicate rubidium as a tracer for wood burning.

   Rubidium has been found to be emitted by woodburning processes at trace levels (0.1 % of PM mass) and two citations have been added supporting this.
   "Elements and organic compounds associated with wood combustion: rubidium (Kleeman et al., 1999; Svane et al., 2005), potassium, levoglucosan, mannosan and galactosan (Urban et al., 2012) were constantly ranked highly in terms of importance (Figure 5)."
   Rubidium is redox-active and other studies have documented a high correlation between rubidium and OP.[5]

4. Figure 5, the color points are quite scattered, however, no discussion on uncertainties at all.

   The coloured points in this figure represent the importance ranking of an independent variable for a site. Because importance is a rank, there is no associated uncertainty to the discrete rank returned by the model. The scatter is a result of site-specific differences.

5. line 218, PM-coarse contained much of OP signal, it would be helpful to provide numbers, ie. % of PM-coarse in total PM

   This point was also raised by Reviewer 2. The text has been edited to include these percentages: "Another key observation from these aggregations was that $PM_{coarse}$, defined as the mass concentration of PM with a size between 2.5 and $10\,\mu m$, contained 50 and 45 % of the $OP_v^{AA}$ and $OP_v^{DTT}$ signal respectively. This was only able to be highlighted because the sampling design included both $PM_{10}$ and $PM_{2.5}$.'"

[4]Weber, S. et al. (2021). Source apportionment of atmospheric $PM_{10}$ Oxidative Potential: synthesis of 15 year-round urban datasets in France. *Atmospheric Chemistry and Physics* 21.14, pp. 11353–11378. URL: https://acp.copernicus.org/articles/21/11353/2021.

[5]Calas, A. et al. (2019). Seasonal Variations and Chemical Predictors of Oxidative Potential (OP) of Particulate Matter (PM), for Seven Urban French Sites. *Atmosphere* 10.11, p. 698. URL: https://www.mdpi.com/2073-4433/10/11/698.

6. line 223, 'lower levels of structure" is confusing. do you mean low levels of spatial and seasonal variation?

   This sentence has been edited with the suggested wording:
   "The DCFH assay based on a fluorescence method showed much lower levels of spatial and seasonal variation when compared to the other two, more established AA and DTT assays where the means ranged between..."

7. line 120, typo in 25 ug L-1? Should be ug mL-1 instead?

   The unit for extraction concentration was indeed incorrect. This has been corrected to $25\,\mu g\,mL^{-1}$.

8. line 273, typo in DFCH

   Corrected.

**Other changes**

1. The units used for $OP_v^{DCFH}$ has been clearly stated in the text and the relevant plot axis labels.

2. All new commands have been removed from the LaTeX source file.